# Intimate Partner Violence Screening Instruments: A Protocol for a COSMIN-Based Systematic Review

**DOI:** 10.3390/ijerph20021541

**Published:** 2023-01-14

**Authors:** Yanjia Li, Keyi Zhou, Siyuan Tang, Jiarui Chen

**Affiliations:** 1Xiangya School of Nursing, Central South University, Changsha 410013, China; 2Xiangya Center for Evidence-Based Nursing Practice & Healthcare Innovation: A JBI Affiliated Group, Changsha 410008, China

**Keywords:** intimate partner violence, screening instruments, COSMIN

## Abstract

Intimate partner violence (IPV) is a major public health problem resulting in a significant impediment to equal participation, quality of life, and personal, social, and economic development. At present, a variety of screening instruments for IPV have emerged in developed countries, and some of them have been adapted to the language and culture of different countries, such as Hurt, Insult, Threaten, Scream (HITS) and the Abuse Assessment Screen (AAS). The selection of the most appropriate IPV screening instrument for the target population and context from among those instruments has become difficult for researchers when intending to start screening. Therefore, a systemic review of IPV screening instruments is needed. This protocol describes a COSMIN-based systematic review of the measurement properties of these instruments. The aims of the systematic review are to (1) evaluate the methodological quality of studies on the measurement properties including the validity, reliability, and internal consistency of these IPV screening instruments, and (2) provide suggestions for relevant researchers in their local context for using the IPV screening instruments.

## 1. Introduction

The World Health Organization (WHO) has defined Intimate Partner Violence (IPV) as “any intimate relationship that results in physical, psychological or sexual harm to the person in the relationship, including physical, sexual violence, emotional-psychological abuse and deprivation of liberty [1]”. To understand IPV, it is important to have a clear understanding of the word “intimate partner”; in IPV, the intimate partner refers to heterosexual, homosexual, and bisexual relationships [2], regardless of age, gender, or marital status [3]. Thus, IPV is different from the word “domestic violence”. Domestic violence refers to abuse against children, adults, and the elderly, including economic, physical, sexual, emotional, and psychological abuse [4]. Therefore, we can conclude that intimate partner violence (IPV) from a current or former spouse or partner does not only include domestic violence, which means domestic violence is one type of IPV.

Intimate partner violence (IPV) is a major public health issue and human rights threat worldwide [5,6,7,8]. The World Health Organization reported [9] that 27% of women aged 15–49 have experienced physical and/or sexual violence from their partners, and 13% of women have experienced such events in the past 12 months. In 2018, a study showed [10] that the prevalence of psychological violence, physical violence, and sexual violence in China was 77.7%, 40.2%, and 11%, respectively, and 52% of people experienced two or three types of violence during their lifetime. Although most victims of IPV are women and most perpetrators are men [1], men may suffer from IPV in homosexual and heterosexual relationships [1,9]. More and more studies have found that the prevalence of IPV among gay, bisexual, and other men who have sex with men is at the same or higher level than that of heterosexual women [11,12,13,14]. A study in Nigeria investigated [14] the prevalence of IPV among gay, bisexual, and other men who have sex with men and found that the prevalence rates of experiencing emotional, physical, sexual, monitoring, and control behaviors were 45%, 31%, 20%, 55% and 22% respectively. A domestic study [15] found that 26.5% of men who have sex with men have at least one IPV in verbal, physical and sexual violence. Given the high prevalence worldwide, IPV deserves our attention regardless of sex, age, or marital status.

IPV is related to a series of adverse health outcomes, including physical, sexual, reproductive health, and mental health problems, such as depression, anxiety, drug abuse, and post-traumatic stress disorder [16,17]. Research has found that the overall depression rate of women experiencing IPV was 65.8%, of which 69.5% had experienced physical violence and 75.8% had experienced sexual violence, respectively [10]. Many efforts have been made to reduce the effects of IPV. One of the most important findings was that studies have confirmed that early screening for people without signs or symptoms of abuse can find the abusive behavior that has not been disclosed separately, and provide opportunities for intervention to reduce future abusive behavior and the short-term and long-term adverse health consequences [18,19]. Therefore, it is very necessary to screen for IPV, while at the same time, it is also necessary to screen according to the characteristics of the participants, such as gender, gender identity, and sexual identity.

Many researchers have described the IPV screening instruments in the context in which the instruments were originally developed. However, these tools may be developed for different people and contexts; for example, Hurt, Insult, Threaten, Scream (HITS) was developed for both female [20] and male [21] patients in predominantly Hispanic clinical settings, and the Abuse Assessment Screen (AAS) [22] was mainly developed to screen abused women during pregnancy and has been used for abused pregnant and non-pregnant African-American, Hispanic, and white women in health and prenatal clinics and emergency departments. As there are many IPV screening tools, the applicable population and context are different, and the quality of the methodology is also uneven, which makes it difficult problem for a researcher to choose an IPV screening tool with a high methodological quality that is suitable for their target context. Recently, in order to select the best outcome measurement instrument, the Amsterdam Public Health research institute, VU University Medical Center, Amsterdam, the Netherlands has developed COnsensus-based Standards for the selection of health Measurement INstruments (COSMIN) methodology for systematic reviews of Patient-Reported Outcome Measures (PROMs) and the COSMIN Risk of Bias checklist for systematic reviews of PROMs, which provides a process to help researchers evaluate the measurement properties and quality of instruments. In this study, we will conduct a systematic review of the methodological quality of IPV screening tools. At the same time, we also want to provide some suggestions for researchers who want to use the IPV screening instruments in their context.

Therefore, the overall objective of this systematic review is to strictly describe, appraise and summarize the screening instruments for IPV according to the COSMIN guideline on a systematic review of PROMs [23,24,25]. The specific aims of the drafted systematic review are: (1) to identify and describe all existing and validated instruments for screening IPV; (2) to assess the methodological quality of studies on the measurement properties of these instruments; and (3) to evaluate and compare the characteristics of these instruments.

This systematic review answers the following five questions: (1) What are existing screening instruments for people who are experiencing or have experienced IPV? (2) What are the characteristics of these screening instruments? (3) What is the methodological quality of studies on the measurement properties of these screening instruments? (4) What are the measurement properties, internal consistency, construct validity and reliability of these screening instruments? (5) What are the differences and similarities among these screening instruments?

## 2. Methods

Based on the COSMIN guideline for systematic reviews of PROMs, we adapted the ‘objectives’ section of the Preferred Reporting Items for Systematic reviews and Meta-Analyses (PRISMA) 2020 27-item checklist [26,27] by changing ‘Population(s), Interventions, Comparators and Outcomes’ into ‘Construct, Populations, Type of Instruments and Measurement Properties’. We registered the systematic review in the International Prospective Register of Systematic Reviews (PROSPERO), CRD42022365247. We are conducting this systematic review based on the COSMIN guideline for systematic review PROMs and will report it following the 2020 27-item checklist. The process for conducting a systematic review of PROMs is shown in Figure 1.

### 2.1. Inclusion and Exclusion Criteria of the Studies

Studies will be included if they: (1) report screening IPV instruments designed for people suffering from or who have suffered from IPV; (2) describe the processes of development and/or evaluation of one or more measurement properties for the eligible instrument(s); (3) have full-text availability; and (4) are published in English.

Studies will be excluded if they: (1) do not report the complete data; (2) are not primary studies (e.g., discussion papers, letters, and editorials) or case studies; (3) use the screening instruments only for outcome measurement; or (4) are repeated publications of studies.

### 2.2. Search Strategy

Using an appropriate and comprehensive search strategy that combines medical subject headings and free test words, we are searching five databases, including PubMed, EMBASE, Web of Science, ProQuest Dissertations & Theses Global (PQDT Global), and EBSCO PBSC (Psychology Behavioral Sciences Collection). The search time ends on 10 November 2022. The main search concepts are the consequences of IPV (Construct), people who suffer from IPV (Population), PROMs (Type of instrument(s)), and measurement properties. Therefore, our strategy for this search is to use ‘intimate partner violence’ (medical subject headings), ‘mass screening’ (medical subject headings), ‘multiphasic screening’ (medical subject headings), and the subordinate word and entry terms, ‘IPV’ (acronym), ‘screening instrument*’ (free test word), ‘screening tool*’ (free test word) using ‘AND’ and ‘OR’ operators to identify relevant articles to enter the search. An example of the search strategies used in PubMed is presented in Table 1. The reference lists of included articles are also being manually searched to identify any potentially relevant studies. Our team members developed the search strategy through constant discussion to ultimately determine the final search plan with the help of librarians. We are also periodically reviewing updates to database articles, and, if needed, we will re-implement the above search strategy to identify new eligible studies and will promptly update the content of this systematic review.

### 2.3. Study Screening

Endnote [28] is being used to manage reference filtering. First, we are using Endnote to identify and remove duplicates, and we are grouping articles from different databases and implementing manual filtering. After the initial screening, the title, abstract, and full-text articles will be independently reviewed and screened by two researchers based on the inclusion and exclusion criteria to identify any potentially relevant study. Any disagreements will be resolved through discussion. If necessary, a third researcher will be sought. The reference lists of all qualified articles will also be screened according to the inclusion and exclusion criteria mentioned earlier. The process of research screening is shown in Figure 2.

### 2.4. Data Extraction

According to the purpose of this study, we will formulate a data extraction table through group discussion. The data extraction table contents will include (1) general information about the included studies (including publication year, journal, DOI number, and first author name); (2) the basic characteristics of the identified instruments (including the name of the instrument, developer(s) or developed year, number of items, score range, original language and available language(s), application context, targeted people and instrument structure); (3) the consequences of the measurement properties of the identified instruments (including the validity, reliability, and internal consistency); and (4) the feasibility, the feasibility does not belong to a measurement property in the evaluation of the instruments, which refers to the ease of use of the PROM in its intended context, and mainly depends on whether the IPV screening tool can be accepted by the research object, the time to complete the scale, the cost, and the quality of completion. At the same time, we will compare the feasibility of all PROMS according to these points to help choose the most appropriate PROM when encountering PROMs belonging to the same category. The feasibility results will be presented in a table. The two researchers will extract the data separately, and finally, the extracted data will be determined on the basis of group discussion. A third researcher will review the extracted data and resolve any differences between the two researchers in the process of extracting data.

### 2.5. Quality Appraisal

The quality evaluation for the included studies and instruments will be conducted independently by two researchers using the corresponding COSMIN box according to the COSMIN guidelines for systematic reviews. If there is a lack of consensus, it will be discussed with the third researcher. We will evaluate the quality of each instrument separately with respect to each measurement property using the corresponding COSMIN box [23]. To ascertain the overall rating of the quality of each instrument with respect to a measurement property, we will take the lowest rating of any standard in the box, i.e., the “worst score counts” principle will be employed. Finally, each instrument will be graded as being of high, moderate, low, or very low quality.

### 2.6. Data Synthesis

In the first step, four measurement properties of each of the included instruments will be evaluated: validity, reliability, internal consistency, and feasibility. Taking the evaluation of internal consistency as an example, the manner in which internal consistency is to be evaluated according to the COSMIN guidelines is presented in Table 2. The COSMIN Risk of Bias checklist [24] will be used when starting to evaluate the methodological quality of each measurement property for each of the included instrument. In this step, we will use an Excel spreadsheet with the filename “COSMIN Box Score” to determine the evaluation scores. The methodological quality results will be presented as a table.

In the second step, the updated COSMIN quality criteria for good measurement properties [23] will be used to assess the consequences of all of the measurement properties of each of the included instruments.

In the third step, we will statistically pool or qualitatively summarize the consequences of the measurement properties of each included instrument from different studies. The pooled or summarized results will be presented in a table in the review. If necessary, we will use a meta-analysis approach (weighted means and 95% CI). The pooled or summarized results for each measurement property for each PROM will be rated against the updated COSMIN quality criteria for good measurement properties.

Finally, we will summarize the evidence and grade the quality of the evidence by using the GRADE [23] (Grading of Recommendations Assessment, Development, and Evaluation) approach developed by the COSMIN guidelines for the systematic review of PROMs. The quality of the evidence refers to the confidence in whether the pooled or summarized results are trustworthy. The quality of the evidence for measurement properties will be rated as high, moderate, low, and very low evidence, with four levels in total according to the GRADE approach. Next, we will describe the feasibility and select a PROM to formulate recommendations.

### 2.7. Formulate Recommendations

To provide researchers in relevant fields with evidence-based and public suggestions on the use of IPV screening instruments, we will follow the recommendations in the COSMIN guidelines [23] to classify the PROMs into the following three categories:(1)If a PROM has enough evidence of content validity (at any level) and at least enough quality evidence of internal consistency, we will classify it as category “A”.(2)If a PROM is neither A nor C, we will classify it as category “B”.(3)If a PROM has high-quality evidence that a measurement property is insufficient, we will classify it as category “C”.

We recommend using PROMs classified as category “A”, and the results obtained by using these PROMs are reliable. We can recommend using PROMs classified as category “B”, but further research is needed to evaluate the quality of these PROMs. However, PROMs classified as category “C” are not recommended. If only PROMs classified as category “B” are found during the systematic review, we can only temporarily recommend the PROM with the best content validity until further and better evidence can be provided.

In the systematic review, we will describe in detail why PROMs are classified into a certain category. At the same time, to standardize the measurement results, we will make suggestions based on one of the most appropriate PROMs. We will not only provide suggestions based on the measurement properties, but also provide suggestions based on the feasibility.

## 3. Conclusions

As far as we know, this will be the first systematic review of IPV screening instruments for people who are experiencing or have experienced IPV based on the PRISMA and COSMIN guidelines. This review will identify, describe, appraise, and compare all of the included instruments. We will use the COSMIN guidelines for the systematic review of PROMs to evaluate the methodological quality of all of the included studies on the measurement properties of all of the included instruments and the psychometric properties of all of the included instruments. Through these findings, we will be able to formulate a suggestion for the usage of the existing qualified target screening instruments, which could help medical workers, psychological practitioners, and community workers better screen IPV victims. This systematic review may support future research, facilitate screening for future researchers before the intervention, and provide suggestions and support for future researchers to improve and further develop these screening instruments.

## Figures and Tables

**Figure 1 ijerph-20-01541-f001:**
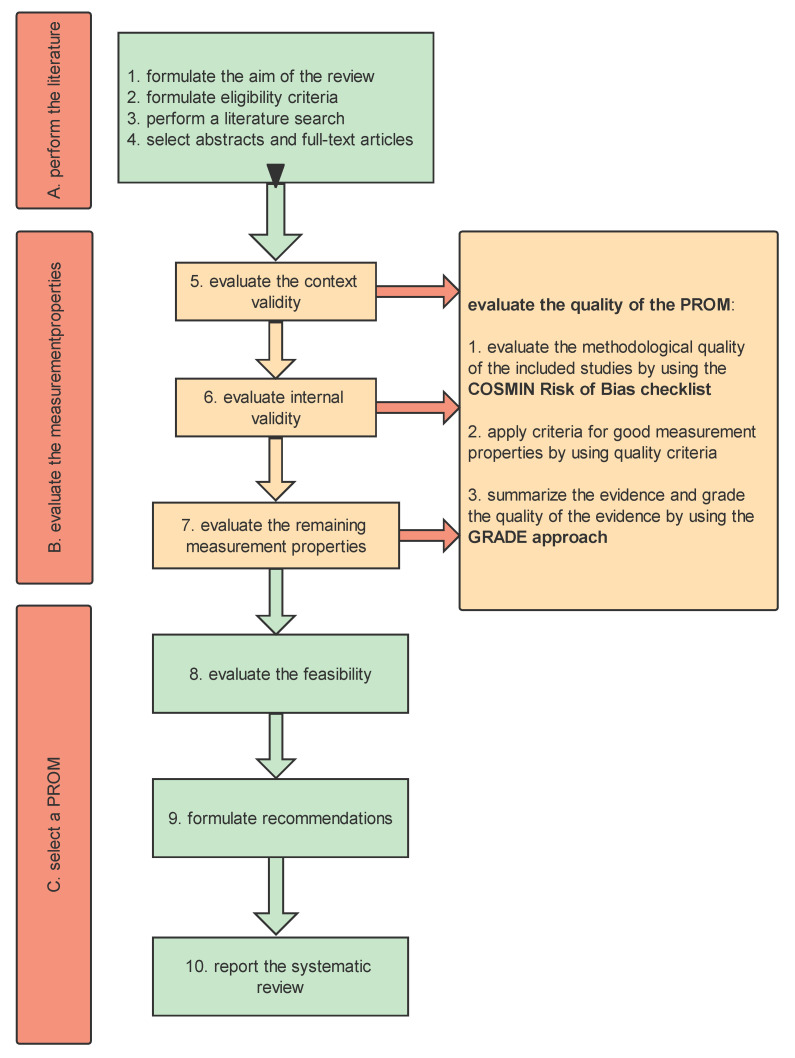
Ten steps for conducting a systematic review of PROMs.

**Figure 2 ijerph-20-01541-f002:**
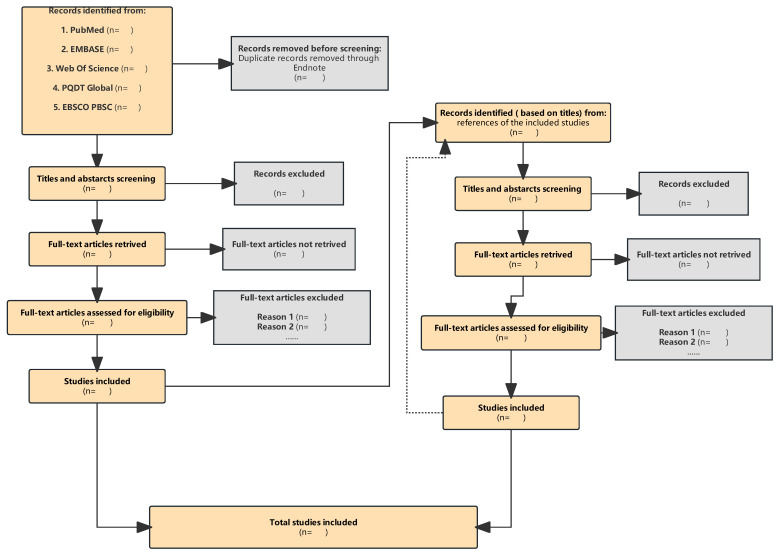
The process of research screening.

**Table 1 ijerph-20-01541-t001:** An example of the search strategies used in PubMed.

#	Search Phrase
**#1**	Intimate partner violence [Mesh] OR Spouse Abuse [Mesh]
**#2**	Spouse Abuse [T/A] OR Intimate partner violence [T/A] OR IPV [T/A] OR Partner Violence, Intimate [T/A] OR Violence, Intimate Partner [T/A] OR Intimate Partner Abuse [T/A] OR Abuse, Intimate Partner [T/A] OR Partner Abuse, Intimate [T/A] OR Dating Violence [T/A] OR Violence, Dating [T/A]
**#3**	Mass Screening [Mesh] OR Multiphasic Screening [Mesh]
**#4**	Screening instrument*[T/A] OR Screening tool*[T/A]
**#5**	Mass Screening*[T/A] OR Screening*, Mass [T/A] OR Screening*[T/A]
**#6**	Multiphasic Screening*[T/A] OR Screening*, Multiphasic [T/A] OR Automated Multiphasic Health Testing [T/A]
**#7**	Screened [T/A] OR detect [T/A] OR detected [T/A] OR detection [T/A]
**#8**	(#1) OR (#2)
**#9**	(#3) OR (#4) OR (#5) OR (#6) OR (#7)
**#10**	(#8) AND (#9)

**Table 2 ijerph-20-01541-t002:** Updated criteria for good measurement properties—internal consistency.

Internal consistency	sufficient	At least low evidence for sufficient structural validity and Cronbach’s alpha(s) ≥ 0.70
	indeterminate	Criteria for “At least low evidence for sufficient structural validity” not met
	insufficient	At least low evidence for sufficient structural validity and Cronbach’s alpha(s) < 0.70

## Data Availability

Not applicable.

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
