# Peer review of "Intimate Partner Violence Screening Instruments: A Protocol for a COSMIN-Based Systematic Review"

_ijerph, 2023, doi:10.3390/ijerph20021541_

Round 1
Reviewer 1 Report
IJERPH-2028211 “Intimate partner violence screening instruments: A protocol for a COSMIN-based systematic review.”
The authors provide a protocol for a systematic review of IPV screening instruments. I agree that IPV is a critical area of study given its widespread impacts on health and wellbeing for adults globally. I appreciate the authors’ intent to provide information and recommendations for researchers and practitioners who might use these instruments.
However, I have concerns about the proposed systematic review. The authors note that their review will be “the first systematic review of screening IPV instruments…based on PRISMA and COSMIN guidelines” and “will identify, describe, appraise, and compare all included instruments.” There have been several comprehensive prior reviews/reports of IPV screeners (see citations below), which have described validity, reliability, and recommendations for use, and it is not clear to me how this systematic review will differ from those prior reports. The protocol does not provide a rationale for why this review is needed, indicate what will be different/novel compared to prior reviews, and also does not provide information on how many new screeners there are to evaluate beyond those that are included in the prior reviews. Given these issues, at this time I cannot make a recommendation for publication.
Additional comments:
· The authors note the importance of also studying IPV among men and individuals who identify as sexual minorities, but do not mention if screeners will be evaluated based on particular sample characteristics, such as sex, gender identity, or sexual identity.
· The authors note that “…there have been some IPV screening tools in different countries and languages.” Will this review focus on language or country in some way, or what screeners seem particularly useful in certain contexts? This was not clear.
· I do not agree with the characterization of the instruments in lines 58-65. For example, they state “…[HITS] can be used for female patients in predominantly Hispanic clinical settings and male patients in clinical settings.” This is not accurate. This particular measure has been used in a variety of samples, including samples of diverse female Veterans across various settings, not simply clinical settings. It is not clear to me what the authors are trying to suggest in this section.
· 2.4 Data Extraction – what is meant by “structure” of the instrument? What is meant by “feasibility?”
· 2.6 Data synthesis – why would you “statistically pool…measurement properties of each included instrument”? If the goal, as stated by the authors, is to describe and compare across instruments, what value would be gained by pooling reliability statistics?
· Figure 1 is blurry and difficult to read, words appear to be missing in boxes
· Figure 2 is also difficult to read
Citations:
Basile KC, Hertz MF, Back SE. Intimate Partner Violence and Sexual Violence Victimization Assessment Instruments for Use in Healthcare Settings: Version 1. Atlanta (GA): Centers for Disease Control and Prevention, National Center for Injury Prevention and Control; 2007
Feltner C, Wallace I, Berkman N, et al. Screening for Intimate Partner Violence, Elder Abuse, and Abuse of Vulnerable Adults: Evidence Report and Systematic Review for the US Preventive Services Task Force. JAMA. 2018;320(16):1688–1701. doi:10.1001/jama.2018.13212
